# Medical Costs and Economic Impact of Hyperkalemia in a Cohort of Heart Failure Patients with Reduced Ejection Fraction

**DOI:** 10.3390/jcm14010058

**Published:** 2024-12-26

**Authors:** Andrea López-López, Margarita Regueiro-Abel, Emilio Paredes-Galán, Charigan Abou Johk-Casas, José María Vieitez-Flórez, Juliana Elices-Teja, Jorge Armesto-Rivas, Raúl Franco-Gutiérrez, Ramón Ríos-Vázquez, Carlos González-Juanatey

**Affiliations:** 1Cardiology Department, Hospital Universitario Lucus Augusti, 27003 Lugo, Spain; andrea.lopez.lopez@sergas.es (A.L.-L.); margarita.regueiro.abel@sergas.es (M.R.-A.); charigan.abou.johk.casas@sergas.es (C.A.J.-C.); jose.maria.vietez.florez@sergas.es (J.M.V.-F.); juliana.elices.teja@sergas.es (J.E.-T.); jorge.armesto.rivas@sergas.es (J.A.-R.); ramon.rios.vazquez@sergas.es (R.R.-V.); 2CardioHULA Research Group, Instituto de Investigación Sanitaria de Santiago de Compostela IDIS, 27003 Lugo, Spain; 3pInvestiga, Moaña, 36950 Pontevedra, Spain; emilio@pinvestiga.com; 4Cardiology Department, Hospital Universitario de A Coruña, 15006 A Coruña, Spain; raul.franco.gutierrez@sergas.es

**Keywords:** hyperkalemia, costs, economic impact, heart failure, reduced ejection fraction

## Abstract

**Background/Objectives**: Hyperkalemia is a common electrolyte disorder in patients with heart failure and reduced ejection fraction (HFrEF). Renin-angiotensin-aldosterone system inhibitors (RAASi) have been shown to improve survival and decrease hospitalization rates, although they may increase the serum potassium levels. Hyperkalemia has significant clinical and economic implications, and is associated with increased healthcare resource utilization. The objective of the study was to analyze the management of hyperkalemia and the associated medical costs in a cohort of patients with HFrEF. **Methods**: An observational, longitudinal, retrospective, single-center retrospective study was conducted in patients with HFrEF who started follow-up in a heart failure unit between 2010 and 2021. **Results**: The study population consisted of 1181 patients followed-up on for 64.6 ± 38.8 months. During follow-up, 11,059 control visits were conducted, documenting 438 episodes of hyperkalemia in 262 patients (22.2%). Of the hyperkalemia episodes, 3.0% required assistance in the Emergency Department, 1.4% required hospitalization, and only 0.2% required admission to the Intensive Care Unit. No episode required renal replacement therapy. Reduction or withdrawal of RAASi was necessary in 69.9% of the hyperkalemia episodes. The total cost of the 438 hyperkalemia episodes was €89,178.82; the expense during the first year accounted for 48.8% of the total cost. **Conclusions**: Hyperkalemia is frequent in patients with HFrEF. It is often accompanied by a modification of treatment with RAASi. Hyperkalemia generates substantial costs in terms of healthcare resources and medical care, especially during the first year.

## 1. Introduction

Hyperkalemia is a frequent electrolyte disorder in patients with heart failure and reduced ejection fraction (HFrEF) [1]. It is defined as an increase in plasma potassium concentration to >5.0 mEq/L, although the clinical guides consider cut-off points of up to 5.5 mEq/L to be acceptable and secure, with no need to modify neurohormonal treatment [2]. Higher plasma potassium values, especially >6 mEq/L, are associated with changes in cell action potential, as well as in cardiac conduction and excitability [3], which could result in increased arrhythmias and mortality [4,5].

Although the results of the different studies are very heterogeneous, the prevalence of hyperkalemia is estimated to be about 3% in the general population [6]. These figures can reach 20% in patients with chronic heart failure [7], mainly because of the concomitant presence of comorbidities such as chronic kidney disease and diabetes mellitus, as well as neurohormonal treatment [8,9].

In patients with HFrEF, the use of renin-angiotensin-aldosterone system inhibitors (RAASi), which include the combination of neprilysin and angiotensin II receptor inhibitors (ARNi), angiotensin converting enzyme inhibitors (ACEi), or angiotensin II receptors inhibitors (ARBi), together with mineralocorticoid receptor antagonists (MRAs), at the maximum tolerated doses, has been shown to improve survival and reduce hospital admission rates [2,10,11]. In this regard, biomarkers such as N-terminal pro B-type natriuretic peptide (NT-proBNP) can help guide the prognosis in these patients [12]. However, these drugs also contribute to increasing serum potassium levels through different mechanisms of action [13], thus limiting their use or resulting in a decrease in dose or even suspension of treatment, with the consequent loss of prognostic benefit [14,15]. In a recent study, the existence of hyperkalemia hindered the reaching of aimed doses in up to 9.3% of the patients jointly treated with ACEi/ARBs and MRAs and was moreover responsible for 8.9% of the contraindications or intolerances to some of these drugs [1].

The prevention and treatment of hyperkalemia requires a multidisciplinary approach. In the case of life-threatening hyperkalemia, the main aim is to avoid fatal arrhythmias by adopting measures that allow stabilization of the myocardial cell membrane, the transport of extracellular potassium into the cells, and elimination of the cation from the body as quickly as possible. If these measures do not prove effective, hemodialysis may be required in some cases. In the rest of patients, the initial measures consist of a low-potassium diet, the withdrawal of potassium supplements and nephrotoxic drugs, and the start or titration of loop or thiazide diuretics. The prolonged use of binding agents such as ion exchange resins is not advised, due to the possible associated adverse effects and lack of safety and efficacy studies [16,17,18,19].

In many cases, the absence of adequate treatment alternatives for potassium control ultimately leads to the suspension or dose reduction of RAASi drugs, which can worsen the patient prognosis and increase mortality. According to the clinical guides, if short term suspension of these drugs is required, it should be minimized and the treatment should be carefully reintroduced as soon as possible, with close monitoring of the serum potassium levels [20]. New potassium binding agents, such as patiromer or sodium zirconium cyclosilicate, constitute novel therapeutic options allowing the maintenance of RAASi drugs in those patients that need them despite the development of hyperkalemia [2]. Pivotal studies have demonstrated efficacy and safety in terms of the normalization of elevated potassium levels, the maintenance of normokalemia over time, and the prevention of recurrence of hyperkalemia in these patients. Such drugs therefore can be considered for the treatment of this disorder under certain circumstances [21,22,23].

Hyperkalemia is therefore associated to important clinical and economic consequences, with an increase in the use of healthcare resources, including more visits to the emergency room or specialist care, an increase in hospital admissions, and a rise in mortality [24]. In this context, there is a need to understand the management of hyperkalemia and its impact in terms of the use of RAASi drugs in real clinical practice, and to quantify the associated resources and costs in a cohort of patients with HFrEF.

The principal objective of the present study was to evaluate the management of hyperkalemia in a cohort of patients with HFrEF, and its clinical and economic impact. Evaluation was performed surrounding modifications in RAASi treatment, the number of visits to the emergency room, hospitalizations, and the need for additional consultations relating to the correction of the electrolyte imbalance. In addition, the economic impact and medical costs associated with the management of hyperkalemia were examined.

## 2. Material and Methods

### 2.1. Study Design and Screening Criteria

An observational, longitudinal, retrospective, single-center retrospective study was made of consecutive patients starting follow-up in the Specialized Heart Failure Unit of the Department of Cardiology of a secondary hospital between January 2010 and December 2021.

The study included patients with HFrEF defined according to the criteria of the 2021 clinical practice guide of the European Society of Cardiology (ESC) [2]. All patients were Caucasian, over 18 years of age, and were referred from the Hospitalization Unit or other Cardiology clinics. None of them presented decompensated heart failure.

Loss to follow-up was defined as an absence of contact with the patient in the 6 months prior to the start of the statistical analysis.

### 2.2. Intervention and Data Collection

The collection of data referred to demographic, clinical, laboratory test, electrocardiographic, and echocardiographic parameters, as well as information referring to treatment, hospital admissions, visits to the emergency room, or successive controls by the specialist. This was carried out by the doctors of the Unit based on reviews of the electronic case histories of the patients. The study variables were entered into a statistical database intended for the purpose of the study.

Patient management followed the current heart failure guidelines at the time of the patient’s in the clinic.

Hyperkalemia was defined as a serum potassium concentration of >5.5 mEq/L. The plasma potassium levels were determined before each patient visit, with none being determined during hospital admission. Hyperkalemia was considered to be present at the first measurement meeting the diagnostic criterion.

General recommendations regarding a low-potassium diet for patients with HFrEF, particularly those with chronic kidney disease, are provided during the first visit to the Unit.

The corresponding cost of consumables and laboratory materials was assigned to each laboratory test, along with the cost of the extraction visit and the cost of the telephone contact (to report the result to the patient and modify treatment, program another test, or confirm a follow-up visit).

All the patients presented an electrocardiogram (ECG) prior to consultation.

The follow-up starting date for this study was that of the first outpatient visit to our Heart Failure Unit.

The economic costs were calculated based on Spanish Royal Decree 56/2014 of 30 April [25], establishing the healthcare service rates for centers dependent upon the Galician Health Service and public healthcare foundations of Galicia. The assessment of hospital costs, whether in the Emergency Service, inpatient hospitalization, or in the Intensive Care Unit, includes the administered medications, the tests performed, as well as the hospital fees during the hospital stay.

### 2.3. Ethical Particulars

The study was performed following the Good Clinical Practice (GCP) guidelines, ethical principles, and the Spanish legal specifications referring to research in force at the time of the investigation. Approval was obtained from the local Regional Research Ethics Committee (protocol code 2022/343; date of approval 21 September 2022) before starting the study.

### 2.4. Statistical Analysis

Qualitative variables were reported as frequencies and percentages. The normality of quantitative variables was explored with Kolmogorov’s test. Variables with normal distribution were reported as the mean and standard deviation (SD); variables without normal distribution were reported as median and interquartile range (IQR).

Quantification was made of the costs of the emergency room visits, stays in the hospital ward, admission to the Intensive Care Unit (ICU), and follow-up laboratory tests. The costs of the consumables and laboratory materials (€9.91), the cost of the extraction visit (€14.52), and the cost of the subsequent telephone contact (€34.52) were quantified. The cost of each of these items is shown in the results section according to Spanish Royal Decree 56/30 April 2014 [25].

The SPSS statistical package version 19 (IBM, Armonk, NY, USA) was used throughout.

## 3. Results

### 3.1. Baseline and Clinical Characteristics of the Study Sample

Of the patients followed-up on in our Heart Failure Unit, 1181 met the criteria for participation in this study. The mean patient age was 66.6 years, and 22.7% were females. Mean creatinine clearance (MDRD) was 72.0 ± 24.7 mL/min/1.73 m^2^, and 31.3% of the patients presented creatinine clearance (MDRD) < 60 mL/min/1.73 m^2^. The rest of the baseline characteristics of the study sample are shown in Table 1.

The mean duration of follow-up was 64.6 ± 38.8 months (median 56; IQR 30.5–97.5), during which a total of 11,059 control visits took place. During the follow-up, there were 374 exits (with a mean survival of 54.2 ± 35.1 months and a median of 48.3 months). There were 24 losses to follow-up (2.0%). These visits documented 438 hyperkalemia episodes in 262 patients (Figure 1, flow chart). Of these 262 patients, 163 (62.0%) presented a single episode, while 99 (38.0%) suffered more than one hyperkalemia episode.

The mean potassium concentration of the 438 hyperkalemia episodes was 5.76 ± 0.30 mEq/L.

### 3.2. Management of Hyperkalemia and Need for Medical Care

The pharmacological treatment was modified in 341 hyperkalemia episodes (77.9%), and one of the essential drug groups was reduced or suspended in 306 episodes (69.9%). In the 35 hyperkalemia episodes in which treatment was modified but without reducing the essential drug groups, the loop diuretic was modified in 16 episodes, potassium binders were added in 17, and the binder drug doses were increased on 2 occasions.

With regard to the essential drug groups, ACEi/ARBi or ARNi were withdrawn in 65 episodes (14.8% of the hyperkalemia episodes), and could be reintroduced in 48 cases, with a median of 11.5 days (IQR 6.0–50.3). In turn, the MRAs were withdrawn in 129 hyperkalemia episodes and could be reintroduced in only 31.5% of the cases, with a median of 111 days (IQR 11–1191). Resins or potassium binders were started in 10.6% of the hyperkalemia episodes.

Emergency room care in hospital was required in 13 episodes (3.0%). Of these cases, six (1.4%) required hospital ward admission and one required admission to the ICU. In no case was renal replacement therapy needed due to hyperkalemia. The mean stay in the hospital ward was 7.2 ± 3.2 days.

In the days after the visit on which hyperkalemia was detected, a total of 953 laboratory tests were performed on an ambulatory basis, followed by additional telephone contacts to monitor the evolution of the potassium levels.

### 3.3. Hyperkalemia Episodes During the First Year of Follow-Up

A total of 196 hyperkalemia episodes (44.7%) occurred during the first year (Figure 2). Of these episodes, eight required emergency care (4.1%) and three (1.5%) required conventional hospital ward admission. No admissions to the ICU or use of renal replacement therapy proved necessary. Neurohormonal treatment was modified in 164 of the 196 hyperkalemia episodes (83.7%) during the first year of follow-up. In 22.0% of the cases, ACEi/ARBi or ARNi were suspended, and in 48.8% of the cases, MRAs were withdrawn. Resins or potassium binders were prescribed in 11.0% of the patients.

### 3.4. Resource Utilization and Associated Costs

An average of 4.21 ± 3.71 laboratory tests were made in the period between the visit prior to the hyperkalemia episode and the episode.

After determining the maximum hyperkalemia value, an average of 2.18 ± 1.30 subsequent laboratory tests were made until normokalemia was confirmed.

With regard to the costs, Table 2 shows that at the end of follow-up, the cost related to conventional hospital ward stays was €25,918.56, while the cost of ICU stays was €2284.95. The cost of care in the emergency room was €4700.65. The cost associated with the laboratory tests and posterior clinical monitoring totaled €56,274.65. Over an average follow-up of 64 months, the calculated total cost related to healthcare resource utilization and the clinical management of hyperkalemia was €89,178.83.

During the first year of follow-up, the total cost corresponding to the 196 recorded hyperkalemia episodes was €43,508.17, representing 48.8% of the hyperkalemia-related costs over 4.5 years of follow-up.

## 4. Discussion

Our study has two main findings. On one hand, most of the costs derived from the hyperkalemia episodes in a cohort of patients with HFrEF were concentrated in the course of the first year. On the other hand, the main cost was associated with the laboratory tests and successive visits to monitor the evolution of plasma potassium and introduce opportune treatment optimizations. The number of admissions to hospital was low.

In our study the incidence of hyperkalemia during follow-up of the patients with HFrEF was high (22.0%) and consistent with the data reported by other studies [7,26]. As our group has previously reported [26], a relevant finding is that there was an increased incidence of hyperkalemia in the first year of follow-up, which could be explained by the fact that it is in the initial months following the patient’s referral to the Heart Failure Unit that the progressive titration of neurohormonal treatment with RAAS happens. In our case, at one year of follow-up, 97.2% of the patients were under treatment with ACEi/ARBi/ARNi at higher doses than at the start; 67.3% were receiving MRAs at doses similar to those at the start; and a lesser proportion (55.2%) of subjects were treated with diuretics at lower doses than at the start. This initial increase in the incidence of hyperkalemia caused the costs of the disorder to be concentrated in the first year. In effect, 48.8% of the total cost corresponded to this period.

In addition, half of the visits to the emergency room and hospital ward admissions occurred during the first year of follow-up, which is precisely when the greatest percentage of hyperkalemia episodes was detected. These results contrast with those of other studies [27,28], where greater incidences of hospitalization, emergency room visits, and hospital visits were recorded than in our series. However, it should be noted that these studies did not involve only patients with heart failure.

In the 11,059 visits during follow-up, we detected 438 hyperkalemia episodes in 262 patients, and of these, 40% experienced more than one episode over time. This observation is consistent with the data published by Thomsen et al. [29], who found the recurrence of hyperkalemia to be frequent in patients with heart failure, resulting in an increase in hospital admissions and thus of healthcare resource utilization.

In our sample, the mean potassium concentration was 5.76 ± 0.30 mEq/L. According to the expert opinions included in the KDIGO guides [30], and in the absence of electrocardiographic changes, this potassium level would be regarded as corresponding to mild hyperkalemia. The close monitoring of these patients after the optimization of RAASi treatment, with periodic laboratory tests (4.21 ± 3.71 on average), could explain the early detection of the hyperkalemia episodes. Likewise, the need for fewer subsequent laboratory tests (2.18 ± 1.30 on average) until normokalemia is confirmed could be justified.

In line with the above, the fact that most of the hyperkalemia episodes involved potassium levels of <6 mEq/L could explain why emergency room care was only required in 3.0% of the cases, with admission to the hospital ward in 1.4% and admission to the ICU in only 0.2%. In no case did renal replacement therapy prove necessary. This also explains why the main cost observed in our study was centered on the outpatient monitoring of potassium elevation, with frequent laboratory tests and telephone visits.

At present, the clinical practice guides [2,31,32] recommend different measures in application to patients with chronic hyperkalemia involving potassium levels > 5.5 mEq/L. These include a low-potassium diet, the withdrawal of potassium supplements and nephrotoxic drugs, or the start of non-potassium sparing diuretics. Furthermore, and given the prognostic benefits of RAASi in patients with HFrEF, those patients who develop hyperkalemia could also benefit from potassium-reducing drugs to introduce or increase neurohormonal treatment [2,16].

However, in the same way as previously described in other observational studies [1], the suspension or reduction of RAASi treatment is common practice in the control of hyperkalemia in patients with HFrEF, with a consequent impact in terms of morbidity and mortality. In our study, RAASi treatment was modified (drug reduction or withdrawal) in almost 70% of the hyperkalemia episodes. Specifically, ACEi/ARBi or ARNi were suspended in 65 episodes, and could be reintroduced relatively early during follow-up (after about 10 days) in 73.8% of the cases, once the potassium levels were controlled. In turn, MRAs were withdrawn in 149 hyperkalemia episodes, but could only be reintroduced much later (after >3 months) and in much lower proportion than with ACEi/ARBi or ARNi (in only 31.5% of the cases). Ion exchange resins or potassium binders were prescribed in only 10.6% of the cases. In the case of the ion exchange resins, this low percentage could be explained by the lack of scientific evidence of their efficacy and the possible adverse effects that might result from their prolonged use [19] and which limit their prescription. In the case of the potassium binders, both patiromer [33] and sodium zirconium cyclosilicate [34] have been postulated as new therapeutic alternatives. On one hand, these drugs would make it possible to maintain RAASi in those patients who need them despite hyperkalemia, and on the other hand they could protect these same patients against hyperkalemia recurrence, with a good efficacy, tolerability, and safety profile even over the long term, thus exerting a positive impact upon the clinical course of patients with HFrEF [33,35,36,37]. In effect, the current clinical practice guides [2] for the first time offer different recommendations to facilitate the management of hyperkalemia in patients with HFrEF, combining potassium binders. However, the strict requirements for authorizing their use [38] complicate and largely limit their prescription, as reflected in our study.

As previously commented, hyperkalemia has important clinical consequences, with a very significant economic impact. In our study, the mean cost per assisted hyperkalemia episode at the end of follow-up was €203.60, with a total hyperkalemia episodes cost of €89,178.83. It is important to note that during the first year of follow-up, the total cost was €43,508.17, representing almost half of the total cost at the end of the 5 years of follow-up. In our series it was during the first year when RAASi drug titration took place, with a low prescription of concomitant potassium binders, and thus the largest observed number of hyperkalemia episodes occurred. As a result, it seems logical for this to be the time when most laboratory tests were made and more visits to the emergency room or hospital admissions took place, giving rise to increased healthcare expenditure. Mention also should be made of the important cost of the laboratory test controls and posterior monitoring at the end of follow-up (€56,274.65) in our series, as previously mentioned. Perhaps the close monitoring of these patients, with periodic laboratory tests, allows for the early detection of hyperkalemia episodes that are largely mild, with early treatment modifications, avoiding subsequent emergency care and hospital admissions, and with clear minimization of the costs at this level. A few studies have addressed the economic impact of chronic hyperkalemia, particularly in a cohort of patients with HFrEF [28,39]. However, as in our case, the existing studies all highlight the important added costs involved, due not only to the direct complications of hyperkalemia but also to the modification in neurohormonal treatment, which is associated with an increased number of cardiovascular events, progression of cardiorenal disease, and increased mortality [40]. It is also important to note that patients with HFrEF often present multiple associated comorbidities that can promote disease progression and further complicate the timely implementation of treatment. Optimizing their management requires a multidisciplinary approach, which can also increase healthcare costs [41]. Based on the above, the healthcare cost generated by the management of hyperkalemia in patients with HFrEF appears to clearly exceed the cost derived from the new potassium binders, as is reflected in the current clinical practice guides [2].

## 5. Conclusions

Hyperkalemia is a frequent finding in patients with HFrEF. It is usually accompanied by a modification in RAASi treatment, which results in a loss of prognostic benefit for the patients. Furthermore, hyperkalemia episodes generate substantial costs in terms of healthcare resources and medical care, particularly during the first year. Close clinical monitoring with periodic laboratory tests as well as the sustained and early use of the new binding agents, would allow for the early detection of hyperkalemia episodes and the introduction of timely modifications capable of improving the patient prognosis and reducing healthcare costs.

## 6. Study Limitations

Our study has limitations. Firstly, this is a retrospective observational study, with the inherent limitations this implies. In addition, the definition of hyperkalemia differs from that used in the clinical practice guides and in most previous studies (potassium > 5 mEq/L) [2]. Nevertheless, as previously stated, a potassium concentration of up to 5.5 mEq/L represents the cut-off level up to which neurohormonal treatment can be maintained without requiring modification. Additionally, it ought to be noted that the decision to modify RAASi may be based on other parameters not considered in this analysis, such as the trend in various measurements. Mention also must be made of the low percentage of patients in our cohort treated with sodium-glucose cotransporter 2 inhibitors (SGLT2i), since their use was not indicated in the clinical practice guides in force at the start of the study [20]. Furthermore, the study was designed to determine the management and costs of hyperkalemia, but we did not contemplate the costs derived from progression of the disorder or mortality derived from the modification of neurohormonal treatment. Lastly, it should be noted that the economic cost was estimated for the Galician Health Service (Spain), and cannot be generalized to other Spanish or European health systems.

Despite this, the characterization and long follow-up period of our cohort of patients with HFrEF is one of the strong points of the research. In addition, an extensive evaluation has been made of the economic impact of hyperkalemia in patients with HFrEF, considering different clinical parameters (hospital stay, emergency care visits, and the use of drugs, and the cost associated to each of them).

## Figures and Tables

**Figure 1 jcm-14-00058-f001:**
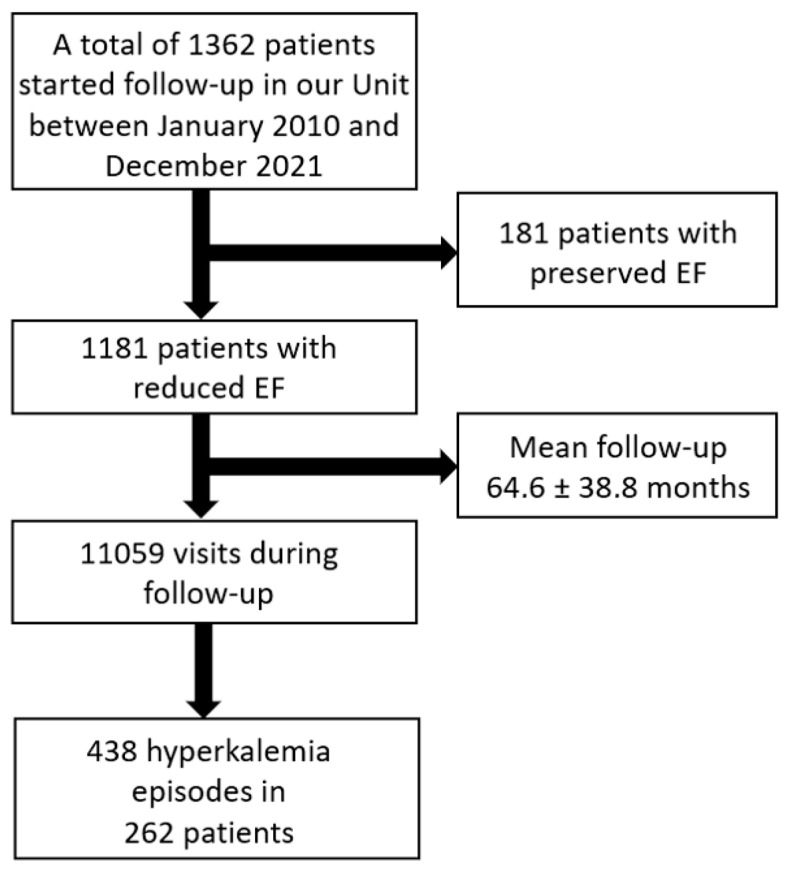
Flow chart.

**Figure 2 jcm-14-00058-f002:**
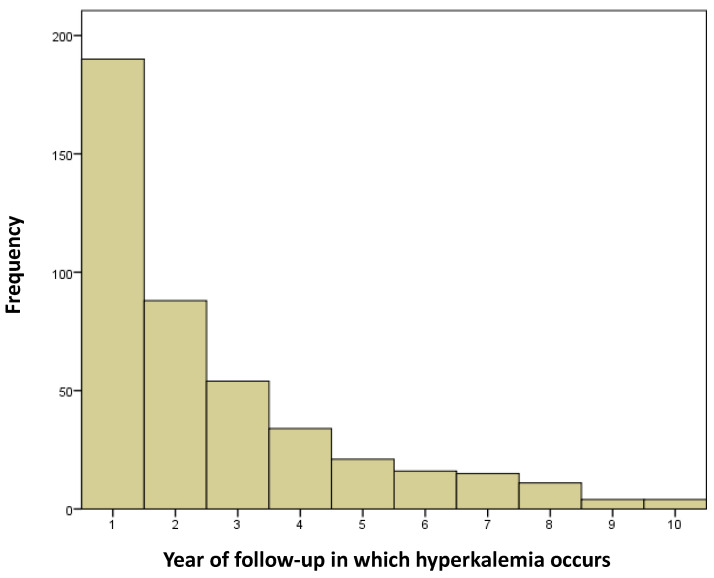
Hyperkalemia episodes over the course of follow-up.

**Table 1 jcm-14-00058-t001:** Baseline characteristics of the study population.

Characteristics		N = 1181
Males, *n* (%)		913 (77.3%)
Age (years)		68 (59–75)
Arterial hypertension, *n* (%)		721 (61.0%)
Diabetes mellitus, *n* (%)		448 (37.9%)
Smoking (ex-smoker and active smoker), *n* (%)		631 (53.5%)
Alcohol abuse (previous and current consumption: >2 alcoholic beverage equivalents per day in males and >1 in females), *n* (%)		494 (41.8%)
Body mass index (kg/m^2^*)*		28.1 (25.5–32.6)
NYHA functional class ≥ 2, *n* (%)		843 (71.4%)
Systolic blood pressure (mmHg)		127.0 (113.3–140.0)
Diastolic blood pressure (mmHg)		69.0 (62.0–77.0)
Primary cause, *n* (%)		
	Ischemic	467 (39.8%)
	Idiopathic	204 (17.4%)
	Alcoholic	101 (8.6%)
	Tachycardia-induced cardiomyopathy	100 (8.5%)
	Hypertensive	88 (7.5%)
	Others	221 (18.2%)
Left ventricular ejection fraction (%)		33.0 (27.0–37.0)
Hematocrit (%)		41.8 (38.5–45.0)
Creatinine clearance, MDRD (mL/min/1.73 m^2^)		72.0 ± 24.7
Creatinine clearance < 60 mL/min/1.73 m^2^, *n* (%)		371 (31.3%)
Basal potassium (mEq/L)		4.63 (4.38–4.94)
Basal potassium > 5.5(mEq/L), *n* (%)		24 (2.0%)
Treatment (first consultation), *n* (%)		
	ARNi/ACEi/ARBi	1149 (97.3%)
	BB	1143 (97.8%)
	MRAs	699 (59.1%)
	SGLT2i	132 (11.2%)
	Loop diuretics	703 (67.3%)
	Potassium supplements	2 (0.2%)
Drug doses at first consultation (over 1, maximum dose according to ESC guidelines)		
	ARNi/ACEi/ARBi	0.5 (0.25–1.0)
	BB	1.0 (0.5–1.0)
	MRAs	0.5 (0.0–0.5)
	Loop diuretics	1.0 (0.0–1.0)
Treatment (at 1 year), *n* (%)		
	ARNi/ACEi/ARBi	955 (97.2%)
	BB	947 (97.8%)
	MRAs SGLT2i	622 (67.3%) 171 (17.4%)
	Loop diuretics	492 (55.2%)
Drug doses at one year (over 1, maximum dose according to ESC guidelines)		
	ARNi/ACEi/ARBi	0.88 ± 0.56
	BB	0.80 ± 0.34
	MRAs Loop diuretics	0.32 ± 0.28 0.63 ± 0.78

Abbreviations: ACEi: angiotensin-converting enzyme inhibitors; ARBi: angiotensin II receptor antagonists; ARNi: neprilysin and angiotensin II receptor inhibitor; SGLT2i: sodium glucose co-transporter-2 inhibitor; BB: beta-blocker; ESC: European Society of Cardiology; MDRD: Modification of diet in renal disease; MRA: mineralocorticoid receptor antagonist; NYHA: New York Heart Association.

**Table 2 jcm-14-00058-t002:** Summary of general costs due to hospital services in patients with hyperkalemia episodes.

Item	Health Service Assigned Cost	Total Number	Total Cost	Number, First Year	Cost, First Year
Emergency care	€361.59	13	€4700.65	8	€2892.72
Stay in ward (days)	€528.95	49	€25,918.56	28	€14,810.6
Stay in ICU (days)	€1142.48	2	€2284.95	0	0
Follow-up laboratory test *	€14.52 + €34.52 + €9.91	953	€56,274.65	437	€25,804.85
Total cost			€89,178.83		€43,508.17

* Follow-up laboratory test includes the costs related to the extraction visit, the material used in testing, and the physician telephone contact. ICU: intensive care unit.

## Data Availability

Data will be made available on reasonable request to the corresponding author.

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
