# Peer review of "Medical Costs and Economic Impact of Hyperkalemia in a Cohort of Heart Failure Patients with Reduced Ejection Fraction"

_jcm, 2024, doi:10.3390/jcm14010058_

Round 1

Reviewer 1 Report

Comments and Suggestions for Authors

1. The cost calculation highlights the cost of emergency care, hospital stays, and laboratory tests; however, the manuscript does not mention the cost of drugs, functional tests, and hospitalization fees. As a result, I could not evaluate whether laboratory tests and successive visits account for the majority of the cost. Please clearly describe what is included and excluded from the cost calculation, and provide reasons for any exclusions.

2. Using only one measurement over 5.5 mEq/l as the diagnostic criterion for hyperkalemia seems unstable. The clinical decision to reduce RAASi could be based on the trend of serum potassium values from multiple measurements. A sensitivity analysis using alternative criteria is recommended.

3. In the conclusion, the authors point out that the use of new potassium-binding agents is limited in Spain due to strict authorization conditions; however, no data presented in the manuscript supports this statement. This statement should be moved to the discussion section. 

Author Response

MAIN CHANGES INTRODUCED.

We have worked diligently for the past ten days to implement all suggested changes and modifications in the best manner possible. We would like to thank the four reviewers for their suggestions, which assisted us in locating errors and significantly enhancing the quality of the article.

We remain at your disposal for any other alterations you deem necessary.

AUTHOR’S RESPONSES TO REVIEWER´S COMMENTS

RV: Reviwers’ comments

AA: Authors’ Answer

REVIWER 1

RV_ Reviwer 1.  Question 1. The cost calculation highlights the cost of emergency care, hospital stays, and laboratory tests; however, the manuscript does not mention the cost of drugs, functional tests, and hospitalization fees. As a result, I could not evaluate whether laboratory tests and successive visits account for the majority of the cost. Please clearly describe what is included and excluded from the cost calculation, and provide reasons for any exclusions

.

AA:  First of all, thank you for the comment. Indeed, multiple questions can arise regarding the costs included in the analysis, considering the different healthcare systems that exist around the word. In our study, the economic costs were calculated based on Royal Decree 56/2014, of April 30, which establishes the rates for healthcare services for centers dependent on the Galician Health Service and public health foundations in Galicia (Spain). It is important to note that the assessment of hospital costs, whether in the Emergency Service, inpatient hospitalization, or in the Intensive Care Unit, includes the administered medications, the tests performed, as well as the hospital fees during the hospital stay. We added this clarification in the methods section to avoid any possible confusion as suggested (page 6).

The assessment of hospital costs, whether in the Emergency Service, inpatient hospitalization, or in the Intensive Care Unit, includes the administered medications, the tests performed, as well as the hospital fees during the hospital stay”.

RV_ Reviwer 1.  Question 2: Using only one measurement over 5.5 mEq/l as the diagnostic criterion for hyperkalemia seems unstable. The clinical decision to reduce RAASi could be based on the trend of serum potassium values from multiple measurements. A sensitivity analysis using alternative criteria is recommended

  1. Thank you for the appreciation. Considering the high number of consultations analyzed (11059 in total) and the prolonged follow-up period (11 years of data collection), we opted for a definition of hyperkalemia that was simple, easily reproducible, and reflected in recent clinical practice guidelines (K>5.5mEq/l is the cutoff point at which changes in neurohormonal treatment are indicated). It is true that the decision to modify RAASi may be based on other parameters not considered in this analysis, such as the trend in various measurements. Thus, a sensitivity analysis was not performed. We included this point in limitations section. (page 16).

Additionally, it ought to be noted that the decision to modify RAASi may be based on other parameters not considered in this analysis, such as the trend in various measurements”.

RV_ Reviwer 1.  Question 3:  3. In the conclusion, the authors point out that the use of new potassium-binding agents is limited in Spain due to strict authorization conditions; however, no data presented in the manuscript supports this statement. This statement should be moved to the discussion section. 

  1. Thank you very much for the comment. It is indeed mentioned in the discussion, and it is also accompanied by bibliography that justifies these authorization conditions. We highlight it in the text (page 14) and move to the discussion section as suggested.

However, the strict requirements for authorizing their use (41) complicate and largely limit their prescription, as reflected in our study”.

Reviewer 2 Report

Comments and Suggestions for Authors

Thank you for the opportunity to review your manuscript entitled " Medical Costs and Economic Impact of Hyperkalemia in a Cohort of Heart Failure Patients with Reduced Ejection Fraction".

Abstract, title and references.
The aim of the study is clear. The title is informative and relevant. The references are relevant, recent, and referenced correctly.
In recent decades, two very important biomarkers have emerged in various areas of cardiology, particularly in terms of diagnosis, treatment, and prognosis, i.e., high-sensitivity troponin T (hs-TnT) and N-terminal pro B-type natriuretic peptide ( NT-proBNP), whose use is now part of daily practice of every cardiologist (1). It is worth supplementing the information on the usefulness of NtproBNP and Troponin T in the prognosis of patients with heart failure (1)

Introduction.
It is clear what is already known about this topic. The research question is clearly outlined.

Methods.
The process of subject selection is clear. The variables are defined and measured appropriately. The study methods are valid and reliable. There is enough detail in order to replicate the study.

Results.
Quantitative data should be expressed as the mean and standard deviation for normally distributed variables and as the median and interquartile range for not normally distributed variables. You can use the Kolmogorov-Smirnov test to prove the data for normal distribution.

Discussion.
The results are discussed from multiple angles and placed into context without being overinterpreted. The conclusions answer the aims of the study. The conclusions supported by references and results. The limitations of the study are opportunities to inform future research.
Overall. The study design was appropriate to answer the aim. The manuscript is well written and a stimulus for the readership.

Minor revisions:

Did the study notice a correlation between the level of Troponin T or NT-proBNP and the cost of patient care?

Reference:

1.       Doi: 10.33963/v.phj.99553

Author Response

MAIN CHANGES INTRODUCED.

We have worked diligently for the past ten days to implement all suggested changes and modifications in the best manner possible. We would like to thank the four reviewers for their suggestions, which assisted us in locating errors and significantly enhancing the quality of the article.

We remain at your disposal for any other alterations you deem necessary.

AUTHOR’S RESPONSES TO REVIEWER´S  COMMENTS

RV: Reviwers’ comments

AA: Authors’ Answer

REVIWER 2

RV_ Reviwer 2.  Question 1. In recent decades, two very important biomarkers have emerged in various areas of cardiology, particularly in terms of diagnosis, treatment, and prognosis, i.e., high-sensitivity troponin T (hs-TnT) and N-terminal pro B-type natriuretic peptide ( NT-proBNP), whose use is now part of daily practice of every cardiologist (1). It is worth supplementing the information on the usefulness of NtproBNP and Troponin T in the prognosis of patients with heart failure (1).

AA: Thank you very much for the provided information. We completely agree on the diagnostic and prognostic value of both biomarkers in patients with HFrEF. We have added this information in the introduction section (page 3) and included the reference, as suggested.

“In this regard, biomarkers such as N-terminal pro B-type natriuretic peptide (NT-proBNP) can help guide the prognosis in these patients (12).”

  1. Duchnowski P, Åšmigielski W. Usefulness of myocardial damage biomarkers in predicting cardiogenic shock in patients undergoing heart valve surgery. Kardiol Pol. 2024;82(4):423-426.

RV_ Reviwer 2.  Question 2. Quantitative data should be expressed as the mean and standard deviation for normally distributed variables and as the median and interquartile range for not normally distributed variables. You can use the Kolmogorov-Smirnov test to prove the data for normal distribution.

  1. Thank you for the appreciation. We carried out the indicated changes. Qualitative variables were reported as frequencies and percentages. Normality of quantitative variables were explored with Kolmogorov´s test. Variables with normal distribution were reported as the mean and standard deviation (SD) meanwhile variables without normal distribution were reported as median and interquartile range (IQR) (page 6 and table 1).

“Qualitative variables were reported as frequencies and percentages. Normality of quantitative variables were explored with Kolmogorov´s test. Variables with normal distribution were reported as the mean and standard deviation (SD) meanwhile variables without normal distribution were reported as median and interquartile range (IQR).”

RV_ Reviwer 2.  Question 3:.Did the study notice a correlation between the level of Troponin T or NT-proBNP and the cost of patient care?

  1. Thank you very much for the suggestion to evaluate the correlation between troponin T and NT-proBNP and the cost of hyperkalemia. It is indeed a very interesting issue. Given that the data collection presented in this manuscript began over 10 years ago (2011), troponin T levels were not recorded. Additionally, at that time, the systematic use of NT-proBNP in the follow-up of patients with HFrEF was not widespread, and data on this biomarker were also scarce. Thus, of the 438 hyperkalemias, NT-proBNP values were only available for 171. Considering your suggestion, we evaluated whether there was a correlation and found none: p=0.561, estimated using Pearson's correlation. In light of the absence of data and statistical correlation, we have decided not to add it to the document, although we will certainly keep it in mind for future studies.

Reviewer 3 Report

Comments and Suggestions for Authors

Thank you for the opportunity to review this article.

The article presents important insights regarding the frequent occurrence of hyperkalemia in patients with heart failure with reduced ejection fraction. The study, based on a retrospective, single-center analysis, highlights how hyperkalemia affects treatment management and generates significant medical costs. The authors emphasize that while the use of RAASi is effective in improving survival and reducing hospitalizations, it may lead to elevated potassium levels, necessitating treatment adjustments in nearly 70% of cases. The results show that although hyperkalemia rarely requires intensive care unit admission, it involves substantial costs, particularly in the first year of treatment.

It is also important to note that patients with heart failure often suffer from multimorbidity, which further complicates treatment. Optimizing the management of chronic diseases these patients face is also costly and requires comprehensive care. I believe the authors should highlight and address this aspect, as it can significantly impact the overall healthcare costs and treatment outcomes for HFrEF patients. The article provides valuable information about the impact of hyperkalemia on the quality of care for patients with HFrEF and its financial burden, which is a crucial issue both clinically and economically.

Author Response

MAIN CHANGES INTRODUCED.

We have worked diligently for the past ten days to implement all suggested changes and modifications in the best manner possible. We would like to thank the two reviewers for their suggestions, which assisted us in locating errors and significantly enhancing the quality of the article.

We remain at your disposal for any other alterations you deem necessary.

AUTHOR’S RESPONSES TO REVIEWER´S  COMMENTS

RV: Reviwers’ comments

AA: Authors’ Answer

REVIWER 3

RV_ Reviwer 1.  Question 1: It is also important to note that patients with heart failure often suffer from multimorbidity, which further complicates treatment. Optimizing the management of chronic diseases these patients face is also costly and requires comprehensive care. I believe the authors should highlight and address this aspect, as it can significantly impact the overall healthcare costs and treatment outcomes for HFrEF patients. The article provides valuable information about the impact of hyperkalemia on the quality of care for patients with HFrEF and its financial burden, which is a crucial issue both clinically and economically.

AA:  First, I'd like to thank you for your extremely valuable contribution to our research. Indeed, we are completely agree with what you have expressed in your reflection, so we have added it to the discussion (page 15) as suggested.

It is also important to note that patients with HFrEF often present multiple associated comorbidities that can promote disease progression and further complicate the timely implementation of treatment. Optimizing their management requires a multidisciplinary approach, which can also increase healthcare costs”.

Thus, we include a new reference:

  1. Mentz RJ, Kelly JP, von Lueder TG, Voors AA, Lam CS, Cowie MR, Kjeldsen K, Jankowska EA, Atar D, Butler J, Fiuzat M, Zannad F, Pitt B, O'Connor CM. Noncardiac comorbidities in heart failure with reduced versus preserved ejection fraction. J Am Coll Cardiol. 2014 Dec 2;64(21):2281-93.

Reviewer 4 Report

Comments and Suggestions for Authors

It was a pleasure reading your article which had a high quality of English language and presentation. 

I have several comments that I think should be addressed if the paper is to be published:

1) Diet. Nowhere in text has diet been mentioned and it is an important part of good management strategy of elevated potassium levels. I think that adding information on whether dietary instructions were given and what was the compliance as well as potential influence on subsequent dosing adjustment for RAAS inhibitors.

2) Survival and loss of follow-up. It is most certain that a significant number of patients died and a certain number were lost to follow-up. I think that mentioning how many of the patients were lost to follow-up, both from the patients with hyperkalemia and those without, is an important part of the data. 

Author Response

MAIN CHANGES INTRODUCED.

We have worked diligently for the past ten days to implement all suggested changes and modifications in the best manner possible. We would like to thank the two reviewers for their suggestions, which assisted us in locating errors and significantly enhancing the quality of the article.

We remain at your disposal for any other alterations you deem necessary.

AUTHOR’S RESPONSES TO REVIEWER´S COMMENTS

RV: Reviwers’ comments

AA: Authors’ Answer

REVIWER 4

RV_ Reviwer 4.  Question 1: Diet. Nowhere in text has diet been mentioned and it is an important part of good management strategy of elevated potassium levels. I think that adding information on whether dietary instructions were given and what was the compliance as well as potential influence on subsequent dosing adjustment for RAAS inhibitors.

AA:  First, I'd like to thank you for your extremely valuable contribution to our hyperkalemia research. At our center, we have a highly specialized heart failure unit with very close monitoring of our patients. We are two cardiologists and a nurse who are solely devoted to assessing these patients, both working in a highly coordinated, standardized, and protocolized manner in terms of indications, tests, pharmacological titrations, and patient reviews. In our Unit, during the first visit, we provide general recommendations regarding a low-potassium diet to patients with HFrEF, especially aimed at patients with chronic kidney disease; although it is true that most of the recommendations from clinical practice guidelines for managing dietary hyperkalemia have limited scientific evidence and are based on expert opinion. We always try to ensure that the reduction of potassium intake from the diet is integrated with the other nutritional goals of our patients.

To achieve this, we conduct nutritional education to identify foods with higher potassium content (we provide tables and recommend mobile applications), explain appropriate culinary techniques for food preparation to achieve lower potassium content, and teach how to detect hidden sources of potassium. We know that adhering to a low-potassium diet is not easy, and despite following it regularly, we often do not achieve significant reductions in potassium levels; hence, we complement monitoring with periodic analytical controls to carry out the necessary pharmacological adjustments.

We have added this information in intervention and data collection (page 5).

General recommendations regarding a low-potassium diet for patients with HFrEF, particularly those with chronic kidney disease, are provided during the first visit to the Unit.”

RV_ Reviwer 4.  Question 2:  Survival and loss of follow-up. It is most certain that a significant number of patients died and a certain number were lost to follow-up. I think that mentioning how many of the patients were lost to follow-up, both from the patients with hyperkalemia and those without, is an important part of the data. 

  1. Thank you for the appreciation. We have added this information in in study design and screening criteria Baseline (page 5) and clinical characteristics of the study sample (page 9), as suggested.

“Loss to follow-up was defined as the absence of contact with the patient in the 6 months prior to the start of the statistical analysis.”

During the follow-up, there were 374 exits (with a mean survival of 54.2 ± 35.1 months and a median of 48.3 months). There were 24 losses to follow-up (2.0%).”

Round 2

Reviewer 1 Report

Comments and Suggestions for Authors

Thank you for revising the manuscript and addressing my comments.

The incidence curve of hyperkalemia is consistent with my experience in Japanese database studies.

I hope readers will become more aware of the importance of monitoring serum potassium levels.